# Thread- and Capillary Tube-Based Electrodes for the Detection of Glucose and Acetylthiocholine

**DOI:** 10.3390/mi11100920

**Published:** 2020-10-02

**Authors:** Kathryn Uchida, Lauren Duenas, Frank A. Gomez

**Affiliations:** Department of Chemistry and Biochemistry, California State University, Los Angeles, 5151 State University Drive, Los Angeles, CA 90032-8202, USA; kmu@iforty.com (K.U.); lduenas7@calstatela.edu (L.D.)

**Keywords:** acetylthiocholine, diagnostic device, electrochemical sensor, glucose, thread-based electrode

## Abstract

An electrochemical sensor for the detection of glucose and acetylthiocholine (ATC) using thread- and capillary tube-based electrodes is described. Three nylon thread-based electrodes were fabricated by painting pieces of trifurcated nylon thread with conductive inks and threading the electrodes into capillary tubes. Two platforms, one paper-based and the other utilizing bubble wrap, were examined. For the glucose detection, a solution containing glucose oxidase (GOx), potassium ferricyanide (K_3_[Fe(CN)_6_]), and increasing concentrations of glucose (0–20 mM) in phosphate-buffered saline (PBS) was spotted onto the two platforms. Similarly, increasing concentrations of ATC (0–9.84 mg/mL) in acetylcholinesterase (AChE) (0.08 U/mL) and PBS solution were detected. Using cyclic voltammetry (CV), a scanning voltage was applied to yield a graph of voltage applied (V) vs. current output (A). For both platforms, both glucose and ATC concentrations were observed to be linearly proportional to the current output as demonstrated by the increased height of the oxidation peaks. The three-electrode system was simple to fabricate, inexpensive, and could be used for multiple readings.

## 1. Introduction

Diseases such as diabetes mellitus, Alzheimer’s, multiple sclerosis, and myasthenia gravis require the constant monitoring of glucose or acetylthiocholine (ATC). Due to the significant prevalence of such diseases, access to accurate and inexpensive devices to monitor and detect the levels of these species is crucial for patient health.

In 2014, the World Health Organization reported 422 million cases of diabetes mellitus, the prevalence of which has increased from 4.7% in 1980 to 8.5% in 2014 and is expected to continue to increase [1]. Diabetes prevalence has increased even more dramatically in middle- and low-income countries, where access to proper treatment and diagnostics is more limited. In 2016, diabetes was identified as the seventh leading cause of death in the United States, and access to proper treatment and screening can avoid or delay consequences of the disease. Hyperglycemia is a characteristic of diabetes that can affect patients due to the lack of an insulin-triggered response and can cause polyuria, polydipsia, weight loss, and blurred vision. Long-term effects of uncontrolled diabetes include retinopathy, nephropathy, peripheral neuropathy, autonomic neuropathy, and an increased risk of cardiovascular disease [2]. For many patients, the regular monitoring of glucose concentrations in blood or urine is a critical part of managing the disease.

The method of glucose detection used in this study uses the oxidation reaction itself to quantify the amount of glucose present in the solution. The biological range of glucose in healthy adults can range from 4.1 to 11.0 mM, although blood glucose concentration typically remains below 6.7 mM [3,4,5]. However, those with hypoglycemia or hyperglycemia may experience levels beyond this range [5]. To ensure the thread-based sensor’s efficiency in detecting glucose within a biological range, glucose concentrations ranging from 0 to 20 mM were measured in this study. The oxidation of glucose by the enzyme glucose oxidase (GOx) and mediated using potassium ferricyanide (K_3_[Fe(CN)_6_]) can be observed and characterized.

The detection of acetylcholinesterase (AChE) and acetylthiocholine (ATC) are of relevance to the studies that examined neurological diseases such as Alzehimer’s, multiple sclerosis, Parkinson’s, dementia, and myasthenia gravis [6]. Neurotransmitters such as acetylcholine (ACh) are emitted into synapses by neurons that target an adjacent neuron or other types of cell. ACh is used to transmit messages between brain nerve cells and is released to activate muscle contraction. Once the signal has been sent, ACh is hydrolyzed by the enzyme acetylcholinesterase (AChE) to form choline, thus terminating the nerve signal. Acetylthiocholine (ATC) is structurally similar to ACh and acts as a substrate for the same enzyme. ATC is traditionally used as the substrate when measuring AChE activity due to its ability to react with 5,5’-dithiobis-2-nitrobenzoic acid (DTNB) and produce a colored product that can be quantified. Although the electrochemical method used in this study can be used for the detection of either ATC or ACh, ATC was chosen for this study in order to remain consistent with the more traditional colorimetric method of characterizing AChE. ATC in this study is oxidized by AChE to form thiocholine, which is then oxidized to form a disulfide. The oxidation of thiocholine is measured, allowing the concentration of ATC to be determined. AChE concentration was kept constant (0.08 U/mL) in order to electrochemically analyze ATC concentrations (0–9.84 mg/mL).

Lapses in neurotransmitter responses may present in neurodegenerative diseases such as myasthenia gravis, in which the immune system develops a response against ACh receptors, preventing ACh binding and subsequent muscle contraction. Additionally, some toxins, venoms, and insecticides can bind to ACh receptors, preventing ACh from binding [7]. It has also been shown that diseases such as Alzheimer’s can result in ACh drops as significant as 90%. As a result, the detection of both ACh and AChE are important in the diagnosis and management of neurodegenerative disorders [8]. Rapid technology growth has greatly enhanced the scope of real-time health monitoring, facilitating the incorporation of medical diagnostics into patients’ home health care regimens. Microfluidics is an empowering technology for the miniaturization of point-of-care (POC) diagnostic devices, which allows patients to monitor their health quantitatively at home without laboratory analysis and using only minute volumes of sample. Microfluidic-based devices have been implemented into a variety of different applications including cell sorting, enzymatic assays, biotechnology, defense, and POC diagnostics. These platforms are advantageous due to their rapid analysis, low cost, small reagent volumes and size, low power consumption, parallel analysis, high sensitivity and accuracy, and ability to multiplex [9].

The recent integration of thread in microfluidic technology has been fueled by its wide availability, light weight, flexibility, and hydrophilic nature. Additionally, three-dimensional structures can be fabricated by sewing thread onto other materials in different patterns in order to either prevent or facilitate fluid mixing [10]. The gaps between the twisted fibers that make up the thread form capillary channels that allow for fluid to flow up the thread without an external pump system, allowing for easily fabricated and rapid POC diagnostic systems [11]. In this way, thread and thread-paper microfluidics have the potential to provide low-cost, low-volume, and multispecies analysis for monitoring health, environment, and food safety. The electrodes detailed herein, take advantage of the hydrophilic nature and capillary action of nylon thread by ensuring the conductive inks painted onto the surface of the threads permeate and distribute evenly within the threads.

Paper microfluidics is similarly attractive for developing countries and for use in POC settings [12,13,14,15,16,17,18,19,20,21,22,23,24,25,26,27,28,29,30,31,32,33]. Microfluidic chips utilizing paper platforms are advantageous due to their light weight, portability, low cost, disposability, biodegradability, and biocompatibility [34,35]. Specifically, chromatography paper serves as an ideal platform for analyte detection due to its hydrophilic nature, wide availability, high capillary action, and low sample consumption. Hydrophobic barriers can be produced by printing and then heat pressing wax onto the paper, permeating through the paper matrix and restricting the flow of fluid. This method is simple and can produce a wide variety of hydrophobic patterns. These hydrophobic patterns can be produced by hand and via computer applications like Inkscape for consistency in detection site size.

Bubble wrap has been used as a novel platform material for the containment of bioanalyte solution. Its advantages include low cost, wide global availability, light weight, transparency, and wide range of sizes that allow for the containment of various volumes. Bubble wrap can be easily cut with scissors and disposed of after a single use, in addition to being flexible. The bubbles are sterile, thus eliminating the need for expensive sterilization techniques [36]. The spaces within the bubbles can be easily injected using a syringe or a pipet tip, and can act as cuvettes for colorimetric or fluorescent analysis. Previous work has taken advantage of bubble wrap’s transparency in order to use the bubbles as cuvettes for the colorimetric detection of glucose [37].

Various analytes including glucose and ATC can be detected using colorimetric analysis, such as by performing a color-producing reaction on white thread [11]. Paper microfluidics can also be used for colorimetric detection; however, electrochemical sensors provide more quantitative results and are more versatile than colorimetric methods [34]. The detection of different concentrations of analytes has been successful through the use of electrochemical sensors with conductive Ag/AgCl and carbon ink. The sensors consist of a reference electrode, a working electrode, and a counter electrode. The reference electrode, constructed of Ag/AgCl, carries a fixed potential against the electric potential applied to the working electrode, made with carbon ink, when measuring the electron flow from each concentration of analyte [38]. This effective three-electrode system allows the opportunity to incorporate paper and thread-based materials into fabrication. Electrochemical detection provides a reproducible platform to detect the increasing concentrations of glucose and ATC.

Herein, we report the development of a reusable, easily fabricated, and low-cost electrochemical system for the detection of glucose and ATC. The three electrodes (working, reference, and counter) were fabricated from pieces of nylon painted with conductive inks that were then threaded into glass capillary tubes. Two platforms made of wax-printed chromatography paper and bubble wrap were used to contain the solution being measured. The small electrode size and novel microfluidic platforms allow minimal reagent volume and simple execution.

## 2. Materials and Methods

### 2.1. Materials and Equipment

Potassium ferricyanide (K_3_[Fe(CN)_6_]), glucose oxidase (GOx), glucose, disodium phosphate (Na_2_HPO_4_), sodium dihydrogen phosphate (NaH_2_PO_4_), acetylthiocholine iodide, and acetylcholinesterase (AChE) from Electrophorus electricus (electric eel) (137 U/mg) were purchased from Sigma Aldrich. Carbon (St Louis, MI, USA) /graphite ink, silver ink, and silver/silver chloride ink were purchased from Henkel (Dusseldorf, Germany). Artiste #18 White Nylon thread was purchased from Hobby Lobby (Oklahoma City, Okla, USA), and 1 mm and 2 mm capillary tubes were purchased from Wale Apparatus (Hellertown, PA, USA). Circle patterns were created using Inkscape, and black wax was printed onto Whatman Grade 1 Chromatography paper using a Xerox ColorQube 8580. A 0.1 M (pH = 7.0) phosphate-buffered saline (PBS) solution was prepared by mixing disodium phosphate (144.98 mL; 0.2 M) and sodium dihydrogen phosphate (105.02 mL; 0.2 M), and diluting to 0.1 M in 500 mL. The pH of the buffer was then adjusted to pH = 7.0.

### 2.2. Thread-Based Electrode Fabrication

The thread electrodes were fabricated from a piece of nylon thread that was unwound and separated into three thinner nylon threads. The nylon pieces were then heat-pressed for two min at 350 °F to make them flatter and easier to paint. To make the counter electrode, the thread was painted with Ag ink and cured at 120 °C for 15 min, and then painted with a layer of carbon/graphite ink and cured at 120 °C for 10 min. The electrode was inserted into a 2 mm capillary tube so that one end aligned with the end of the tube while the other with the end of the thread remained exposed. One end of the capillary tube was dipped into carbon ink so that a flat surface of conductive ink on the opening of the capillary tube was exposed in order to hold the electrode in place and prevent an analyte solution from traveling up the tube during detection. This allowed for the surface area of the electrode to be controlled and minimized. The electrode was cured at 120 °C for 30 min. The working electrode was fabricated by painting a piece of thread with Ag ink and curing at 120 °C for 15 min. The electrode was inserted into a 1 mm capillary tube, and the end of the tube was encased with carbon/graphite ink before being cured at 120 °C for 30 min. Lastly, the reference electrode was fabricated by painting nylon with Ag/AgCl ink and curing for 10 min at 80 °C. The thread was inserted into a 1 mm capillary tube, dipped into Ag/AgCl ink, and cured for 30 min at 80 °C. The three-capillary tube-encased electrodes were glued together and connected to the potentiostat by the exposed nylon threads.

### 2.3. Microfluidic Platforms for Bioanalyte Detection

Two platforms were used to minimize both the reaction site and solution volume. First, a black wax circle design was printed onto chromatography paper. Inkscape was used to create a circle design 5 mm in diameter with a 10% color opacity fill and a 100% color opacity border acting as a hydrophobic barrier. This barrier allowed the solution spotted within the circle to stay within a defined reaction site. Once printed, the paper was heat pressed for 120 s at 350 °F. Finally, a layer of opaque black wax was printed on the back of the paper, sealing the back of each circle to prevent solution leakage. The solution (12 µL) was spotted within the wax-printed circle, and the three-electrode system was implemented by submerging the electrode ends in the solution bubble, while the exposed threads on the opposite end were clipped to the potentiostat to run CV (Figure 1a). One circle was used per CV for a given bioanalyte concentration.

The second platform was composed of a sheet of bubble wrap with bubbles 8 mm in diameter. A small hole was cut into the plastic with a syringe, through which solution (30 µL) was pipetted into each bubble. The electrode system was inserted into the bubble while the exposed thread ends were clipped to the potentiostat (Figure 1b). Both the bubble wrap and paper platforms were used for the detection of glucose and ATC.

For the glucose detection, the solution was composed of GOx (10 mg/mL) and (K_3_[Fe(CN)_6_]) (10 mg/mL) in PBS, as well as increasing the concentrations of glucose (0–20 mM). A scanning voltage of range −0.5–0.6 V was used to observe the reduction peak and the oxidation peak at 0.45 V. For ATC detection, PBS solution containing AChE (0.08 U/mL) and a range of ATC concentrations (0–9.84 mg/mL) was used. A scanning voltage of 0–1.2 V was applied, and the current output at the height of the oxidation peak at 0.6 V was measured.

## 3. Results and Discussion

The materials used in the fabrication of the chemical sensors offer many advantages over commercial devices and products used in monitoring patients’ health. Nylon thread is inexpensive, easily accessible, absorptive, and conductive when painted with carbon or Ag/AgCl inks. The thread sensors have the ability to maintain their shape while wicking fluid when applying cyclic voltammetry [37]. For the paper-based platform, the wax-printed ink on the chromatography paper acted as a hydrophobic barrier for the solution to be measured within. This platform was easy to fabricate, required small amounts of solution and did not require other materials (e.g., plastic). For the bubble wrap platform, this material is widely available material and is often used as a protective layer for packaging. In this work, each bubble served as a site for the solutions to mix and for the detection required only sample volumes.

### 3.1. Wax-Printed Circle Platform

Using the paper platform, a range of glucose concentrations (0–20 mM) was examined using GOx and K_3_[Fe(CN)_6_] as mediators via CV. Each paper circle held 12 µL of solution containing 10 mg/mL GOx and 10 mg/mL K_3_[Fe(CN)_6_] and an increasing concentration of glucose. The electrode system was placed so that the electrode detection site was submerged in a puddle of solution and was transferred to a new paper circle for each different glucose concentration. From the CV, a proportional correlation between the current output at the observed oxidation peak and the glucose concentration was observed. The optimal detection potential was determined to be 0.45 V due to the easily differentiated glucose oxidation peaks at that potential. CV was performed for each glucose concentration, using a different paper circle for each CV, and the current output at 0.45 V was measured. The range of glucose concentrations was measured in triplicate (Figure 2a). The current output values for each glucose concentration obtained in triplicate were averaged and plotted against the glucose concentration, and the standard deviation of the triplicate values was calculated (Figure 2b) (R^2^ = 0.998).

The paper platform was then used to examine AChE, which catalyzed the hydrolysis of the neurotransmitter acetylcholine (ACh) and can be indicative of both pesticide exposure and neurodegenerative disorders. ATC, which is structurally similar to ACh, was used as the substrate of the reaction, and a range of concentrations (0–9.84 mg/mL) was examined in the presence of AChE (0.08 U/mL) via CV. Each circle held AChE (12 µL) and a known concentration of ATC. The electrode system was used to run the CV for each circle, and the range of ATC concentrations was measured in triplicate (Figure 3a). The height of the ATC oxidation peak at 0.60 V was determined to be the optimal detection potential, and the average current output at 0.60 V for three experiments was plotted against ATC concentration (R^2^ = 0.984). The standard deviation of the triplicate was calculated for each concentration of ATC and is illustrated by the error bars of the graph (Figure 3b).

The use of a printed paper-based platform for the electrochemical detection of analytes is a novel implementation. The chromatography paper utilized for the paper-based platform acted as a hydrophobic barrier with the wax-printed ink for the solution to be measured on. This platform was easy to fabricate, required small amounts of solution and did not require other materials (e.g., plastic). Compared with earlier work, the detection site of the thread electrodes was significantly smaller in size (cm^2^ scale to ~5 mm^2^) [37,38]. The minimization of the electrode, along with the implementation of microfluidic materials, allowed for a significant decrease in solution volume (from 90 µL in earlier work to 12 µL for the wax-printed circles platform) [37,38]. Despite the decrease in electrode size and solution volume, the results obtained are comparable to the results obtained in this work. The oxidation and reduction peaks did not experience any notable potential shifts with an increase in concentration, and the potential at which the oxidation peak occurred is consistent with similar work [37,38]. Results obtained for glucose detection using the wax-printed circles platform are comparable to previous work and demonstrate consistent increases in the measured current output proportional to increases in glucose concentration (R^2^ = 0.998). Measurements of the ATC concentrations gave similar results, yielding a strong linear correlation between the ATC concentration and oxidation peak height at 0.60 V (R^2^ = 0.984).

The materials used in the fabrication of the chemical sensors offer many advantages over commercial devices and products used in monitoring patients’ health. Nylon thread is inexpensive, easily accessible, absorptive, and conductive when painted with carbon or Ag/AgCl inks. The thread sensors have the ability to maintain their shape while wicking fluid when applying cyclic voltammetry.

### 3.2. Bubble Wrap Platform

The capillary tube-based electrode system was then implemented into the bubble wrap platform. The oxidation of a range of glucose concentrations (0–20 mM) mediated by GOx (10 mg/mL) and K_3_[Fe(CN)_6_] (10 mg/mL) was measured via CV. Each bubble was used to measure one concentration of glucose and contained 30 µL of enzyme, mediator, and glucose. The electrode system was inserted into the bubbles, and the CVs for the range of glucose concentrations were obtained in triplicate. The current output at the height of the oxidation peak at 0.45 V was measured (Figure 4a). Glucose concentration was then plotted linearly against the averaged current output (R^2^ = 0.991) (Figure 4b). Lastly, ATC was measured in the presence of AChE. AChE (0.08 U/mL, 30 µL) and a range of ATC concentrations (0–9.84 mg/mL) were pipetted into the bubbles to obtain triplicate CVs. ATC concentration was plotted against the average current output at the oxidation peak at 0.60 V (R^2^ = 0.982) (Figure 4c,d).

Compared with earlier work, the detection site of the thread electrodes was significantly smaller in size (cm^2^ scale to ~5 mm^2^) [33,34]. The minimization of the electrode, along with the implementation of microfluidic materials, allowed for a significant decrease in solution volume (from 90 µL in earlier work to 12 and 30 µL for the paper and bubble wrap platforms, respectively) [39,40]. Despite the decrease in electrode size and solution volume, the results obtained are comparable to the results obtained in past work and provided reproducible results. The detection of both glucose and ATC yielded a correlation between the respective concentration and current output.

Once glucose concentrations from 0 to 20 mM were obtained, the limit of detection (LOD) for glucose using the three-electrode system was determined. Decreasing concentrations of glucose from 1.000 to 0.025 mM were measured via CV (10 mg/mL GOx, K_3_[Fe(CN)_6_]) and the height of their oxidation peaks at 0.40 V measured and plotted linearly (Figure 5a).

Similarly, low ATC concentrations (0.0025–0.5 mg/mL) were measured via CV to determine the LOD for ATC of the system. The AChE concentration was kept constant at 0.08 U/mL. The oxidation peak height was measured at 0.60 V to observe if the CVs behaved expectedly at lower concentrations (Figure 5b).

The LOD for glucose using the electrode sensor and printed circles platform was found to be 0.05 mM glucose (10 mg/mL GOx and K_3_[Fe(CN)_6_]). Concentrations below 0.05 mM could not be measured with enough precision to consistently yield an expected current output at 0.4 V. The CV obtained for 0.025 mM glucose had a slightly lower current output at 0.4 V than 0 mM of glucose did, which is most likely due to the more visible effects of experimental error due to the small difference in glucose concentration. However, the low LOD demonstrates the high sensitivity of the sensor and its ability to detect glucose concentrations below a typical biological range (4.1–11.0 mM).

The LOD for ATC detection was found to be 0.005 mg/mL ATC (0.08 U/mL AChE). At lower concentrations, the current output did not follow the linear trend in current output at 0.60 V as closely due to the decreased difference in concentration. This falls below the normal biological range of ACh (3100 to 6500 U/L in males and 1800 to 6600 U/L in females) [41]. The low LOD for ATC detection shows that the sensors are sensitive and can precisely detect differences in concentration as low as 0.005 mg/mL.

Bubble wrap as a material is widely available and is often used as a protective layer for packaging. In this platform, each bubble served as a site for each concentration to mix. Results obtained for glucose detection successfully show a correlation between the concentration and current output at 0.45 V (R2 = 0.9908). Similarly, the results for ATC detection exhibited a similar relationship (R2 = 0.9823). The results obtained using the bubble wrap system are comparable to both the results obtained using a beaker of solution and the wax-printed circles platform. Despite the fact that the bubble wrap system used more solution volume (30 µL) than the paper circles system (12 µL), the bubble wrap required no printing or other preparation before use. Both the bubble wrap and wax-printed circles used significantly less solution volume than the beaker (12 µL). Both platforms used widely available, low-cost, disposable materials, and allowed for separate detection sites for each measurement.

## 4. Conclusions

We described a unique electrochemical sensor for the detection of glucose and acetylthiocholine using thread- and capillary tube-based electrodes. Here, nylon thread-based electrodes were fabricated by painting pieces of trifurcated nylon thread with conductive inks and threading the electrodes into small capillary tubes. The incorporation of capillary tubes allowed for the minimization of the electrode detection site and subsequent solution volume. The sensors are simple and easy to fabricate, requiring a few basic components.

The three-electrode system was implemented into two microfluidic platforms; a paper-based platform as well as one utilizing bubble wrap as the reaction vessel were used to detect a range of both glucose and ATC concentrations. Both platforms yielded results comparable to previous works and allowed for small volumes of analyte solution (12 µL and 30 µL). Additionally, the capillary tube-based electrodes are extremely sensitive, as the limit of detection (LOD) for the electrode sensor was found to be 0.05 mM glucose (10 mg/mL GOx and K3[Fe(CN)6]) and 0.005 mg/mL ATC (0.08 mg/mL AChE). The low LODs suggest that these sensors can successfully measure glucose concentrations within a biological range as well as very low concentrations of ATC. Low LODs also show that the sensors are very precise and can detect changes in concentration as small as 0.05 mM and 0.005 mg/mL (glucose and ATC, respectively).

The sensors and platforms described are inexpensive to produce and in large numbers and require minimal reagent volumes. The three-electrode system is reusable and can be easily washed and/or the nylon replaced. Additionally, the platforms required little to no fabrication as well as materials that are widely available, light in weight, low-cost, and disposable. The usage of nylon thread, paper, or bubble wrap for sensor and platform fabrication makes these systems an attractive option for use in resource-limited regions requiring POC diagnostic devices with high throughput needs. Future work is focused on utilizing other textiles, further miniaturizing the devices and broadening the scope of analytes studied.

## Figures and Tables

**Figure 1 micromachines-11-00920-f001:**
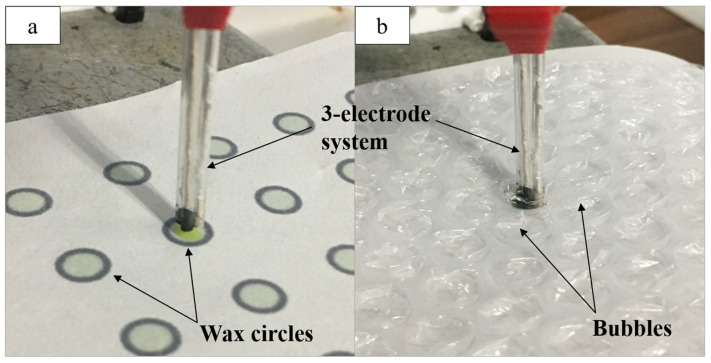
(**a**) The three-electrode system attached to the potentiostat and used for the detection of glucose in solution (12 µL) spotted onto the wax-printed circles; (**b**) the electrode system used for acetylthiocholine (ATC) detection in solution (30 µL) spotted into bubble wrap.

**Figure 2 micromachines-11-00920-f002:**
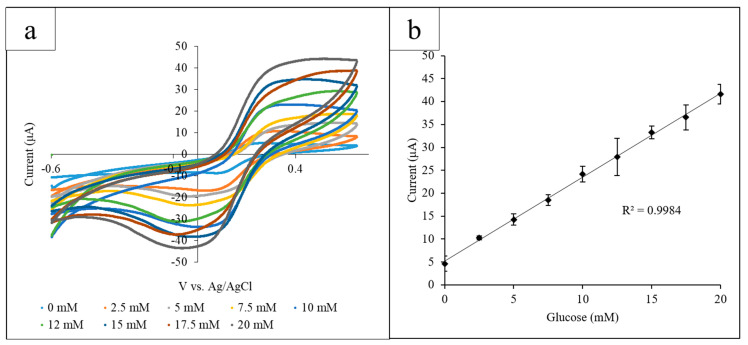
(**a**) Cyclic voltammograms for the increasing glucose concentrations measured using the paper platform in triplicate. The current output at the oxidation peak (0.45 V) increased proportionally with the glucose concentration; (**b**) the calibration curve measuring the current output at 0.45 V vs. glucose concentration. A linear correlation between the current and concentration was found to have an R^2^ value of 0.998.

**Figure 3 micromachines-11-00920-f003:**
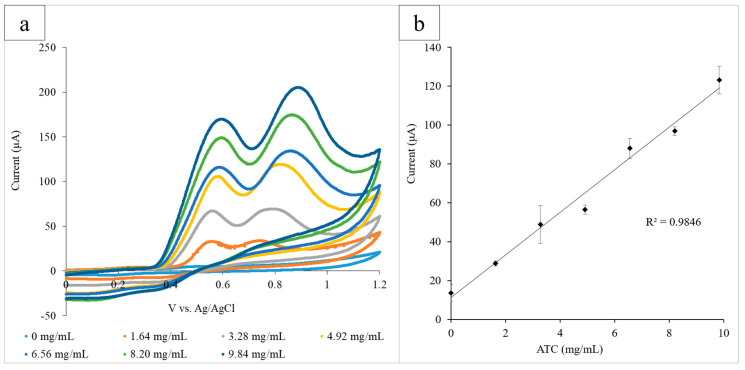
(**a**) Cyclic voltammograms for the increasing ATC concentrations measured using the paper platform in triplicate. The current output at the oxidation peak (0.60 V) increased proportionally with the glucose concentration; (**b**) the calibration curve measuring the current output at 0.60 V vs. ATC concentration. A linear correlation between the current and concentration was found to have an R^2^ value of 0.985.

**Figure 4 micromachines-11-00920-f004:**
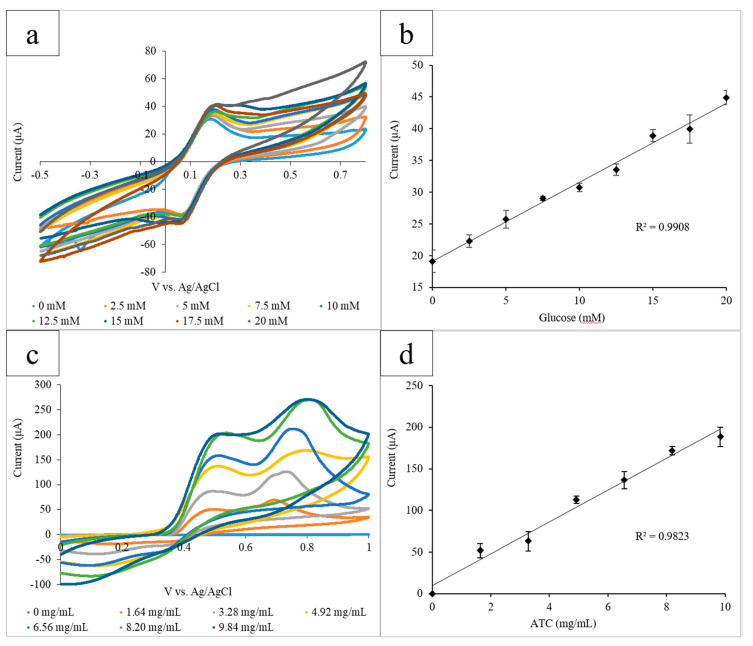
(**a**) Cyclic voltammograms for the increasing glucose concentrations measured using the bubble wrap platform. The current output at the oxidation peak (0.45 V) increased proportionally with the glucose concentration; (**b**) the calibration curve measuring the current output at 0.45 V vs. glucose concentration. A linear correlation between the current and concentration was found to have an R^2^ value of 0.991; (**c**) the cyclic voltammograms for increasing the ATC concentrations measured using the bubble wrap platform. The current output at the oxidation peak (0.60 V) increased proportionally with the ATC concentration; and (**d**) the calibration curve measuring the current output at 0.60 V vs. glucose concentration. A linear correlation between the current and concentration was found to have an R^2^ value of 0.982.

**Figure 5 micromachines-11-00920-f005:**
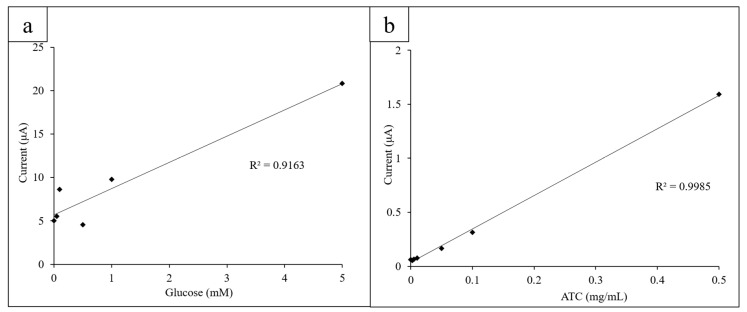
(**a**) Lower concentrations of glucose (0.025–1.000 mM) were measured to determine the LOD for glucose; (**b**) the lower concentrations of ATC (0.0025–0.5 mg/mL) were measured to determine the limit of detection (LOD) for ATC.

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
