# Peer review of "Thread- and Capillary Tube-Based Electrodes for the Detection of Glucose and Acetylthiocholine"

_micromachines, 2020, doi:10.3390/mi11100920_

Round 1

Reviewer 1 Report

In this work, an electrochemical sensor for the detection of glucose and acetylthiocholine (ATC) was reported. The main findings of the manuscript are that the authors ‘developed an electrochemical system for the detection of glucose and bubble wrap was used to contain the solution’. However, the technique described in this manuscript for electrode fabrication was reported by the same authors (in this paper 10.1002/elps.201800348). Moreover, the method of the proposed sensor based on glucose oxidase and potassium ferricyanide is well known. Furthermore, the use of bubble wrap for the containment of the analyte solution was reported previously (10.1016/j.jab.2016.05.003). So, the applied methodology and material in this manuscript is similar to that reported previously and does not contain new and original contributions. Therefore, I am not recommending this paper for publication in Micromachines.

Reviewer 2 Report

The paper entitled “Thread- and Capillary Tube-Based Electrodes for the Detection of Glucose and Acetylthiocholine” by authors Kathryn Uchida, Lauren Dueñas and Frank A. Gomez investigates electrochemical sensing of glucose and acetylthiocholine using thread- and capillary tube-based three-electrode system. They fabricated nylon thread-based electrodes by painting pieces of nylon thread with conductive inks with subsequent threading the electrodes into capillary tubes. The paper-based and bubble wrap platforms were used. The cyclic voltammetry (as a detection method) was used to measure concentration of glucose and acetylthiocholine. Presented sensors are claimed to be inexpensive, simple to fabricate using minimum of material and chemicals.

The paper is scientifically sound and after minor revisions (see below) can be published in Micromachines.

1) Abstract: Despite ‘PBS’ is well known abbreviation please define it in abstract and when used first time in the main text.

2) Ln.48-60, 201: It should be noted more specifically why acetylthiocholine (ATC) and not directly acetylcholine (ACh) is detected in this study.

3) References 16 and 17 are not mentioned in the main text.

4) Fig.2., Fig.3., Fig.4., Fig.5.: Please add thick marks to the x- and y-axes in both panels.

5) Fig. 2b. and 3b.: How the error bars were obtained? Please add some details in figure caption.

6) Please make more clear which values of LOD are for which substrate and platform. It is hard to follow.

7) I have some doubts about LOD values of glucose detection (Fig. 5a). Given the scatter nature of the graph in the range 0-1 mM I cannot imagine that the LOD is only 0.05 mM. Can authors comment on this of give more realistic estimate of LOD.

8) Ln.283: There are mentioned ‘Supplementary Materials’ which are not available and there is no reference to them in the main text. Please fix this.

Reviewer 3 Report

The submitted manuscript describes an ingenious measurement system for the determination of glucose and acetylthiocholine. The authors used readily available materials and accessories that can be found in most chemical laboratories to build a device which allows for sensitive determination of the analytes mentioned earlier. Every stage of this study was described in detail in the experimental section. The obtained results were properly presented and their interpretation does not raise my objections. However, it would be advantageous for a reader if the Authors added a microscopic image of the electrode cross-section and information about the diameter of the nylon fibers used.
